# Instrumented Gait Classification Using Meaningful Features in Patients with Impaired Coordination

**DOI:** 10.3390/s23208410

**Published:** 2023-10-12

**Authors:** Zeus T. Dominguez-Vega, Mariano Bernaldo de Quiros, Jan Willem J. Elting, Deborah A. Sival, Natasha M. Maurits

**Affiliations:** 1Department of Neurology, University Medical Center Groningen, University of Groningen, 9713 GZ Groningen, The Netherlands; z.t.dominguez.vega@umcg.nl (Z.T.D.-V.); m.bernaldo.de.quiros@umcg.nl (M.B.d.Q.); j.w.j.elting@umcg.nl (J.W.J.E.); 2Department of Paediatrics, Beatrix Children’s Hospital, University Medical Center Groningen, University of Groningen, 9713 GZ Groningen, The Netherlands; d.a.sival@umcg.nl

**Keywords:** early onset ataxia, developmental coordination disorder, gait assessment, inertial measurement units, random forest classifier

## Abstract

Early onset ataxia (EOA) and developmental coordination disorder (DCD) both affect cerebellar functioning in children, making the clinical distinction challenging. We here aim to derive meaningful features from quantitative SARA-gait data (i.e., the gait test of the scale for the assessment and rating of ataxia (SARA)) to classify EOA and DCD patients and typically developing (CTRL) children with better explainability than previous classification approaches. We collected data from 18 EOA, 14 DCD and 29 CTRL children, while executing both SARA gait tests. Inertial measurement units were used to acquire movement data, and a gait model was employed to derive meaningful features. We used a random forest classifier on 36 extracted features, leave-one-out-cross-validation and a synthetic oversampling technique to distinguish between the three groups. Classification accuracy, probabilities of classification and feature relevance were obtained. The mean classification accuracy was 62.9% for EOA, 85.5% for DCD and 94.5% for CTRL participants. Overall, the random forest algorithm correctly classified 82.0% of the participants, which was slightly better than clinical assessment (73.0%). The classification resulted in a mean precision of 0.78, mean recall of 0.70 and mean F1 score of 0.74. The most relevant features were related to the range of the hip flexion–extension angle for gait, and to movement variability for tandem gait. Our results suggest that classification, employing features representing different aspects of movement during gait and tandem gait, may provide an insightful tool for the differential diagnoses of EOA, DCD and typically developing children.

## 1. Introduction

Movement coordination is an ability that allows smooth and efficient goal-directed movements involving several body parts. Due to its complexity, a coordinated gait requires the interaction between different body parts, various muscles to maintain balance, sensory inputs and proprioception [1]. The cerebellum is an essential part of the coordination and planning of complex movements. It is well recognized that the cerebellum regulates postural equilibrium and muscle tone by interacting with the brainstem, basal ganglia, and cerebral cortex. The cerebellum also contributes to the cognitive aspects of postural control such as the maintenance of postural verticality and anticipatory postural adjustment [2]. In particular, gait is an ability that starts to develop during the first year of life but maturity is only reached around 11 years of age [3]. The maturity and development of coordinated gait movements are affected by movement and developmental disorders such as dystonia, early onset ataxia (EOA) and developmental coordination disorder (DCD). In this study, we focus on the distinction between EOA, DCD and typically developing (CTRL) children, based on gait.

Ataxia presenting before 25 years of age is denominated as early onset ataxia (EOA). EOA involves lack of coordination, impaired voluntary and goal-directed movements and loss of balance control [4]. Distinguishing clinical features in EOA patients include dysdiadochokinesia, dysmetria, overshoot, impaired gait and posture, intention tremor, oculomotor dysfunction and speech abnormalities [5]. The scale for the assessment and rating of ataxia (SARA) is one of the clinical tools commonly used to assess ataxia. In patients with movement disorders, the SARA was designed to identify and quantify ataxic characteristics [6].

Developmental coordination disorder (DCD) is defined as an impairment in the coordination and execution of motor function which affects the child’s academic or social development, while no underlying neurological diagnosis or intellectual deficits, such as cerebral palsy, muscular dystrophy, visual impairment or intellectual disability, are present [7,8]. Mostly, the features consistent with DCD are identified by the parents when the children present delays in gross or fine motor milestones [8]. Some of the most common symptoms in DCD patients are abnormal coordination, having motor coordination below expectations for their age as well as delays in early motor milestones, such as walking and crawling.

Patients with EOA and DCD may thus present with an overlap in clinical characteristics, hampering the clinical distinction between both [5]. Reliable phenotypic recognition of EOA among other developmental disorders with coordination impairment, such as DCD, in relation to typically developing children is important for selecting the correct diagnostic algorithm, predicting familial recurrence risk, treating the child and/or, in case of typical development of an immature motor system, consolidation of the parents [5]. However, in the context of ambiguous clinical descriptions and in the absence of clear clinical criterion standards, the clinical distinction between mildly initiating EOA features and DCD may be challenging [5]. In the latter study, 5 out of 21 EOA and DCD patients were inhomogeneously phenotyped by three paediatric neurologists. The phenotypic interobserver agreements for the EOA, DCD and central hypotonia (sub)groups (Gwet’s agreement coefficient) were: EOA = 0.801 (*p* < 0.001; substantial); DCD = 0.327 (*p* = 0.037; fair); central hypotonia = 0.415 (*p* = 0.005; moderate). 

Several earlier studies compared the gait of typically developing children with DCD [9,10,11] or with ataxia patients [12,13,14]. However, to our knowledge, few studies have aimed to achieve the clinically more relevant goal of distinguishing between EOA, DCD and the immature motor behaviour of typically developing children, in a single study [15,16]. The reason for this may be that DCD is typically diagnosed by rehabilitation doctors, whereas EOA is typically diagnosed by paediatric neurologists. Since both patient groups are thus examined and treated by different specialists, a direct comparison between the groups may be hampered. In a recent study, DCD and typically developing children were asked to walk barefoot on a motor driven treadmill for 2 min, while their gait was assessed using a 3D motion capture system. The authors found that children with DCD exhibited more complexity in their shank movements and also greater variability in thigh and shank movements compared to age- and gender-matched typically developing participants [9]. In another study, DCD and typically developing children were asked to walk up and down a flat 10-m-long pathway for 1 min, while the movement of their feet and trunk was recorded using motion analysis. The gait pattern of children with DCD was characterised by higher variability and an increased range of movement compared to their healthy peers [17]. In a previous study from our own group, we aimed to distinguish between patients with EOA or DCD and typically developing children [15]. Using inertial measurement units (IMUs), we collected data while the participants walked independently, and derived time and frequency domain-based, as well as statistical, information. Employing this information, we improved the classification performance compared to three expert evaluators. Although we achieved good accuracy, explainability of this classification model was low, i.e., we could not directly relate the distinguishing features to the clinical construct of impaired coordination. Knowing which interpretable features contribute most to classification is informative for doctors assessing these patients, as they may expose new, unstudied parameters distinguishing between the different groups. We have shown before, for upper limb SARA tasks, that informing clinicians of such distinguishing parameters, can help improve phenotypic assessment of EOA and DCD [5].

In the present study, we therefore aimed to provide meaningful features as derived from the quantitative data of the two SARA gait tests to classify EOA and DCD patients and typically developing CTRL children. Our goal was to obtain features with better explainability on the one hand, while achieving similar or even better accuracy, on the other hand. To do so, we developed our features on the clinical premise that impaired gait coordination is characterised by irregularities in movement and loss of smoothness. We extracted gait features from IMU data collected during both SARA gait tests, accordingly. 

## 2. Materials and Methods

The raw data used in this study consist of a data subset recorded for a larger project on the quantification of coordination impairment employing the SARA [15,16,18,19]. We followed the research and integrity codes of the University Medical Center Groningen (UMCG) and the principles of the declaration of Helsinki (2013). The SARA test battery is performed as part of clinical routine, therefore the Medical Ethical Committee of the UMCG provided a waiver for ethical approval of the study. It was also recognized that the attachment of IMUs to the participant’s body is non-invasive. Finally, parents and children 18 years or older were asked to sign informed consent and informed assent was given by minors 12 years or older. 

### 2.1. Participants

We recruited participants between 2014 and 2019. EOA and DCD patients were asked to participate during routine visits to the outpatient clinic of the UMCG. All DCD patients received an independent neurological examination at the outpatient clinic to ensure that no underlying neurological or intellectual deficits were present. Both EOA and DCD patients fulfilled the official inclusion criteria [7,20]. Healthy participants were siblings of the included patients. They were declared to be healthy by their parents and had no other neurological or orthopaedic diagnosis that could theoretically interfere with coordinated motor performance. Intending to balance age, we tried to match the age of the participants in the three groups. None of the included children received medication with known negative side effects on motor coordination. Participants were excluded when they were unable to execute the SARA lower limb gait tests independently. 

### 2.2. Clinical Assessment

We recorded gait and tandem gait SARA test performances using IMUs. During the gait test, participants were asked to walk independently for six meters, turn and walk back the same distance. The tandem gait test consisted of ten independently executed consecutive tandem steps. 

### 2.3. Data Acquisition

Before the performance of both SARA gait tests, the participants were fitted with six lightweight IMUs (51 mm × 34 mm × 14 mm, 23.6 g, Shimmer3, Shimmer, Dublin, Ireland): one IMU was placed on the sternum, another one on the lower back close to the L3 vertebra, two were placed bilaterally halfway each upper leg over the quadriceps and two on the lateral side of the shanks, just above the malleolus. This set-up was chosen to enable various analyses including joint kinematics analysis during SARA motor test performance. For the goal of this study, only the data from a subset of IMUs were used.

Before attachment, the six Shimmer IMUs were calibrated and programmed to record inertial data into embedded SD cards. We configured each IMU to acquire 3D acceleration (±4 G), 3D angular velocity (±500 dps) and 3D magnetic data (±1.9 Ga), at a sampling frequency of 256 Hz. Additionally, we videotaped the start of recording by pressing a button on the multi-charger (Shimmer sensing, Dublin, Ireland) which resets the internal timer of each device so all devices, as well as full SARA gait test execution, are time-synchronised. This allowed later identification and delineation of each of the gait sequences, i.e., forward walking and turning.

### 2.4. Signal Preprocessing

We obtained six files per patient (one per Shimmer device), containing 3D inertial data. Subsequently, we created separate MATLAB (version R2020b, Mathworks, Natick, MA, USA) files for each test (gait and tandem gait) and each of the straight walking sections, before and after turning. Then, we synchronised the data from all devices, using a MATLAB script that employed the starting and ending time of each test to create a global timing for all devices. After synchronisation, we sorted the data from the sensors such that the *x*-axis corresponded to the vertical direction, the *y*-axis to the mediolateral direction and the *z*-axis to the anterior–posterior direction. 

#### 2.4.1. Gait Data Segmentation

After preprocessing, we plotted the mediolateral angular velocity (gyroscope) signals from the shimmers attached to each (right and left) shank. These signals are most representative of gait [21]. By visual inspection, we then selected only those signals that had a similar pattern to that reported by Salarian et al. [21] for gait, for further analysis. 

Based on the algorithm described by Salarian et al. [21], we created a MATLAB script to segment individual steps from each shank gait signal (left and right). First, we used a decimation technique to reduce the noise of the signal by increasing two bits of resolution [22]. Then, we applied a zero-phase 0.5–5 Hz band-pass fourth-order Butterworth filter. This allowed identification of all positive peaks in the signal, which represent the mid-swing events. Subsequently, we identified the adjacent valleys to each peak, where the valley before the peak represents the moment of terminal contact (toe-off) and the valley after the peak represents the moment of initial contact (heel strike). Similar to Salarian et al. [21], we searched for these peaks in time intervals of 0.5 s before and after mid-swing to automatically identify the toe-off and heel strike events, respectively. We only used the gait cycle data for which all three events could be identified. We identified the events for each shank (left and right) separately, but used the information from both shanks to extract temporal and spatial features (Section 2.5). 

#### 2.4.2. Tandem Gait Data Segmentation

Similar to gait segmentation we first plotted the angular velocity recorded from the Shimmer IMUs attached to the shanks (left and right). By visual inspection, we identified the signals that were correctly acquired for further analysis.

For signal segmentation, we followed the same technique as we did for gait (Section 2.4.1). Once each mid-swing event and the corresponding toe-off and heel-strike events were identified, we segmented each gait cycle from toe-off to toe-off instead of from heel-strike to heel-strike as we did in gait. We did this to have more usable tandem gait cycles per patient. 

### 2.5. Feature Extraction

The extraction of gait and tandem gait features employed expert knowledge from clinicians and knowledge about gait characteristics as described in the literature. The latter were derived from IMU-based gait models [23]. We aimed to derive gait features that reflected the gait characteristics clinicians look for during the evaluation of the SARA protocol, which include gait symmetry and regularity, as well as variability of gait performance. The use of such features instead of more general time or frequency domain-based or statistical features will allow for better explainability of the final model. This approach resulted in 36 extracted features. Details of the feature extraction are provided in Appendix A. 

### 2.6. Classification

For further processing, we combined the 36 extracted features (24 for gait and 12 for tandem gait) in a table, where each column represented one of the 36 features. One row of this table contained feature values from one complete gait cycle of gait and one tandem gait cycle, randomly chosen, thereby building a ‘combined movement’ per patient, for classification. Each patient was represented by 10 rows in this table. 

We decided to use a random forest classifier since it performed well in a previous study with a similar application and allows the most relevant features in the model to be identified, thereby contributing to model explainability [18]. For the classifier, we made an implementation pipeline using Python version 3.7 and scikit-learn 0.32. Random forest (RF) is an ensemble learning algorithm that combines several randomized decision trees and aggregates their predictions by averaging. It also returns estimates of feature importance, that we also report here. The latter characteristic of the method provides some insight into the most relevant features for classification which is of importance for the clinical interpretability of the classifier. Similar to a previous study from our group [18], we used 300 trees to reduce the computational cost, and the default Gini index threshold hyperparameter in scikit-learn for our random forest classification, which was repeated 100 times. The average and standard deviation of the classification metrics were used to obtain an estimate of classification performance. 

To deal with the imbalanced number of participants in our dataset (EOA = 18, DCD = 13 and CTRL = 29), we used the ADASYN algorithm, which synthetically oversamples the minority classes (EOA and DCD), to obtain a balanced dataset [24]. In addition, aiming for a robust estimation (avoiding overfitting) and to overcome the relatively small number of samples we implemented leave-one-out cross validation. However, instead of leaving one instance (one row in the feature matrix) out, we took one participant (10 rows) out to be used as a test set, while keeping the remaining participants as a training set. Note that, for each LOOCV iteration, ADASYN was only applied to this training set. We classified each instance from each participant individually and used a majority vote strategy to obtain the classification per participant, per iteration. In case of a tie, the class was randomly determined from the two tied classes. Using this information over all 100 iterations, we also calculated and plotted the probability for each participant of being classified in one of the three classes. The entire procedure is outlined in the flowchart in Figure 1.

Even though accuracy is one of the most used metrics to evaluate classification performance, the interpretation of the results could be misleading if there is imbalance in the number of participants per group (class). In such cases, precision and recall could be used as additional metrics to evaluate the performance of the classifier. In our case, as we have an imbalanced dataset (EOA = 18, DCD = 13 and CTRL = 29) we, therefore, decided to use these metrics to evaluate classifier performance in addition to accuracy. Precision (also known as positive predictive value) summarizes the number of positive predictions that actually belong to the positive class. Precision is simply the ratio of correct positive predictions out of all positive predictions made. In an imbalanced classification problem with more than two classes, precision is calculated as the sum of true positives across all classes divided by the sum of true positives and false positives across all classes. Precision thus quantifies the ratio of correct predictions across both positive classes (EOA and DCD).
(1)$$Precision=\frac{True\;Positive}{True\;Positive+False\;Positive}$$

Recall summarizes how well the positive class was predicted. Recall quantifies the number of positive class predictions in relation to all positive examples in the dataset. Unlike precision that only comments on the correct positive predictions out of all positive predictions, recall indicates missed positive predictions. In an imbalanced classification problem with more than two classes, recall is calculated as the sum of true positives across all classes divided by the sum of true positives and false negatives across all classes.
(2)$$Recall=\frac{True\;Positive}{True\;Positive+False\;Negative}$$

Precision and recall can be combined into a single score that seeks to balance both concerns, called the F-score or the F-measure. The F_1_-measure, which weights precision and recall equally by taking their harmonic mean, is the variant most often used when learning from imbalanced data [25].
(3)$$F_{1}=\frac{2\times Precision\times Recall}{Precision+Recall}$$

## 3. Results

### 3.1. Participants

Data from 79 participants of the three groups (EOA, DCD and CTRL) were collected while performing the SARA gait tests. Data from 60 participants could be included for analysis after application of the exclusion criteria: 18 EOA, 13 DCD and 29 CTRL. Age was not normally distributed, and age ranges varied between groups (EOA: 6–25 years; DCD: 6–16 years; CTRL: 5–25 years). Yet, age did not significantly differ between groups (Kruskal–Wallis test, χ^2^(2) = 3.2404, *p* = 0.198). Table 1 provides a summary of the participant characteristics. Details on individual participants are provided in Table A2 in Appendix B.

### 3.2. Classification

The classifier had a mean accuracy of 82.0% (SD 3.0%) on new data. The confusion matrix (Table 2) was obtained over 100 iterations and the values presented correspond to mean and standard deviation. On a group level, the accuracy was for EOA: mean = 62.9%, SD = 6.3%, for DCD: mean = 85.5%, SD = 5.9% and for CTRL: mean = 94.5%, SD = 2.2%. We also determined the misclassification: of the EOA patients, 16.6 (SD 4.9)% was classified as CTRL and 21.9 (SD 6.2)% was classified as DCD; of the DCD patients, 3.8 (SD 4.7)% was classified as EOA and 13.7 (SD = 3.8)% was classified as CTRL; and finally, of the CTRL participants 4.3 (SD 1.5)% was classified as EOA and 1.2 (SD 1.9)% was classified as DCD.

### 3.3. Feature Importance

Feature importance or the relevance of the feature for the classifier was calculated as the probability of each feature reaching the final node in the ensemble, as described by the decrease of node impurity. According to this metric, there is a combination of features that made it possible to classify between classes. The most relevant features obtained over 100 iterations were those related to variability (the dynamic time warping distance to the mean from the signal of the thigh during tandem gait; the dynamic time warping distance to the mean from the signal of the shank during tandem gait) and range of movement (the range of the hip flexion–extension angle during gait) with mean feature importance of 6.3%, 5.4% and 5.2%. Here, the dynamic time warping distance allows the variability in the acquired signal across gait cycles to be quantified. The first 20 out of 36 features represented 66.3% of the relative importance, as visualized in Figure 2. For detailed information on the features, please see Appendix A. 

### 3.4. Classification Performance

When considering the combination of EOA and DCD as a positive class, the mean precision of the classifier is 0.78 (SD 0.05), with a mean recall of 0.70 (SD 0.05) and mean F_1_ of 0.74 (SD 0.04). For EOA as a positive class, the mean precision is 0.87 (SD 0.05), with a mean recall of 0.61 (SD 0.06) and mean F_1_ of 0.72 (SD 0.05). For DCD as a positive class, the mean precision is 0.72 (SD 0.07), with a mean recall of 0.82 (SD 0.06) and mean F_1_ of 0.77 (SD 0.05). For CTRL as a positive class, the mean precision is 0.85 (SD 0.03), with a mean recall of 0.95 (SD 0.02) and a mean F_1_ of 0.90 (SD 0.02). These and additional measures of classification performance are summarized in Table 3. 

Finally, we calculated the probability for each participant of being classified in any of the three groups. Over the 100 iterations, Figure 3 shows, for all participants, the probability of being classified as EOA (orange), DCD (blue) or CTRL (green). The upper left corner represents a high probability of being classified in the CTRL group. From Figure 3, we can see that 28 out of 29 healthy participants are localized in this corner (probability of being classified as a CTRL participant is >0.5), with one close to the EOA group with a probability of 0.1 of being classified as CTRL. The lower right corner represents a high probability of being classified as a DCD patient. Eleven out of thirteen DCD patients are localised in this corner (probability of being classified as a DCD patient is >0.5) while two DCD patients are close to the CTRL group with a probability of being classified as a CRTL > 0.6. Finally, the lower-left corner represents a high probability of being identified as an EOA patient. Nine of the EOA patients are localised in this corner (probability of being classified as an EOA patient is >0.5), while seven are spread in this probability plot with relatively high probabilities of being classified as a DCD patient or a CTRL participant. Two of the EOA patients are close to the CTRL group, with probabilities of 1.0 of being classified as a CTRL participant.

As can be concluded from Table A2 in Appendix B, in total 10 participants were misclassified; seven EOA patients (five classified as DCD and two as control), two DCD participants (both classified as control) and one control participant that was classified as EOA. Table A2 illustrates that the misclassified participants did not differ in age or sex from the other participants. The gait of the two DCD participants that were misclassified as controls was phenotypically assessed as (almost) normal (three evaluators scored the SARA gait as 0/0/0 and 0/1/0), probably explaining their misclassification. The SARA gait scores—to the extent that we have them available—for the EOA patients that were misclassified are similar to those of correctly classified EOA patients. One EOA patient, that was misclassified as a control, had one of the highest SARA gait scores in this group (3) and walked very broad-based. This is an aspect of movement that we cannot capture with IMUs and may explain why this particular patient was misclassified. The one control participant that was classified as an EOA patient had a normal gait and there is no explanation why this particular participant was misclassified. It should be noted that all participants had to be able to walk independently, leading to relatively mildly affected patients (maximum SARA gait score of 3), in both patient groups, making classification more challenging.

## 4. Discussion

We used IMU data obtained during the execution of both SARA gait tests to extract meaningful features commonly used for gait analysis (e.g., step length and stride velocity) or for representing gait pattern variability (e.g., deviations from the mean). We employed these features for the classification of children with EOA or DCD and typically developing children. We slightly improved classification performance (0.82) when compared to clinical assessment (0.73) [15], but more importantly, we identified the most contributing, meaningful, features, thereby increasing model explainability. We found that features that represent variability in gait are most relevant for classification using a random forest classifier, compared to temporal and spatial features, which contributed less.

The most relevant features used by our random forest classifier were related with the range of the hip flexion–extension angle during gait or with variability of movement during tandem gait. This illustrates that the accurate classification of EOA, DCD and CTRL children depends on the incorporation of features representing variability, from both gait and tandem gait tests. This fits with our a priori hypothesis that abnormal gait is characterised by irregularities in movement and loss of smoothness, and also corresponds with the clinical concept of ataxic gait as visualized in Figure 4. 

As far as we know, there is limited literature on the automatic classification of EOA, DCD and CTRL children based on quantified movement data. Nevertheless, there are some studies aiming to distinguish between these groups. For example, in our previous study using information from gait only, we classified 80.0%, 85.7% and 70.0% of the CTRL, DCD and EOA children, respectively. Overall, that classifier correctly classified 78.4% of the participants [15]. However, in that study we only used statistical features, that provide limited insight in the classification model. In the current study, we correctly classified 94.5%, 85.5% and 61.4% of the CTRL, DCD and EOA children, respectively. Overall, the random forest classifier correctly classified 82.0% of the participants. Not only did we achieve good classification performance, even slightly better than in our previous study, we also now derived knowledge-based features—related to how gait is assessed clinically, incorporating stride length and gait velocity, for example—instead of statistical features, thereby providing meaningful features that enhance the explainability of the classification model.

In the literature, there are some other studies aiming to distinguish between DCD patients and healthy participants using 3D motion capture systems [9,11,16]. In a study aiming to examine the complexity and variability of gait, Rosengren et al. [9] found that DCD patients performed movements with higher complexity, meaning that they had higher variability as well as asymmetry in their movement patterns compared to healthy children. In our study, in line with these findings, we found that the most relevant features used by the classifier to identify between groups are those related to the variability of gait (i.e., distance to the mean trajectory for thigh and shank sensors during tandem gait execution and the range of the hip flexion–extension angle during gait). 

In another study, Woodruff et al. [11], developed an index of walking performance to compare the gait patterns of children with DCD and healthy children. They combined four variables (percentage of cycle at opposite toe off, percentage of cycle of single stance, percentage of cycle at toe off and step length as a percentage of the gait cycle) to create a single index. DCD children showed abnormal gait patterns having larger variance in this index than healthy children. This is in line with our results, again suggesting that variability-based features are more meaningful for DCD distinction from healthy participants. To enhance the interpretability of our results, for the four most relevant features for our classification, we investigated if there was a significant difference between groups, by means of the Kruskal–Wallis H-test. These four features represented variability of movement, two measuring the distance of each gait cycle to the mean (TG_DIS_DTW_MEAN_THIGH, TG_DIS_DTW_MEAN_SHANK), one measuring the range of the angles during each gait cycle (G_RHFE) and one measuring the local curvature from the 3D angular velocity (G_CURV_SHANK). We calculated the median value per most relevant feature and group (see Table 4 and the Appendix A for an explanation of the features). 

For the first three features, the medians are higher for the EOA group, lower for the CTRL group and values for the DCD group are in between, which is expected according to clinical experience. A possible explanation of the increased variability of gait parameters in DCD patients compared to CTRL children could be that DCD patients present increased coactivation, as suggested by Raynor [27], meaning that DCD patients have not developed the same level of muscular organization compared to their healthy peers. Since motor incoordination in DCD patients is clinically associated with cerebellar dysfunction, patients may present with more complex movements, i.e., unskilled movements similar to those observed in typically developing children in the earlier stages of childhood [28].

To the best of our knowledge, this is the first classification study using meaningful features based on the clinical construct of motor incoordination by combining information from the SARA gait and tandem gait tests. Although we obtained a good and insightful classification of EOA, DCD and CTRL children, there are some limitations to the study. First, we used a relatively small dataset (60 participants). However, it should be noted in that respect that, since EOA is a rare disorder, and we wanted to make sure that our labels were as accurate as possible, we carefully identified children with a formal diagnosis of having ataxia, validated by DNA analysis. This limited the number of children with ataxia eligible for our study. Another limitation is the use of inertial measurement units; we are aware that the use of even smaller and lighter sensors could further reduce any experienced restraint during the execution of the gait and tandem gait SARA tests. Hence, participants may not have walked as freely as during regular clinical examinations. However, we avoided pulling the Velcro bands too tight and the Shimmer IMUs only weigh 23.6 g each. Furthermore, we did not take the potential influence of height, cognitive processing, or other disease-independent participant characteristics on gait (variability) features into account. Yet, our results show that features that reflect gait variability during the straight walking part of both SARA gait tasks are most important for classifying the three participant groups. Hence, the potential noise added to these features due to disease-independent participant characteristics apparently does not hamper significant classification. 

In the near future, we aim to include more data from the other SARA subtests, so that we can integrate all information for an improved classification of EOA, DCD and typically developing children. In this study, we aimed to distinguish between EOA, DCD and CTRL participants using meaningful information, deriving features similar to what clinicians look at when phenotypically evaluating patients. We expect that techniques such as deep learning may be helpful in further distinguishing hidden patterns, and can probably provide clinicians with new motor characteristics to consider. However, deep learning techniques typically require ample data, which should then first be collected. 

We further hope that this study motivates other researchers to use wearable sensors for assessment of ataxia and other coordination disorders. Using this technology not only in the clinical environment but also during daily activities could potentially improve the early identification of ataxic signs.

## 5. Conclusions

Combining information from the SARA gait and tandem gait tests and deriving meaningful features for the classification of EOA, DCD and typically developing children resulted in a slightly improved classification performance compared to clinical assessment and increased model explainability. Features representing variability of movement during gait and tandem gait were found to be more relevant for classification between groups compared to temporal and spatial features. These clinically meaningful biomarkers obtained from inertial sensors may contribute to the evaluation of patients with EOA or DCD during gait and thus to the classification of patients with coordination disorders. 

## Figures and Tables

**Figure 1 sensors-23-08410-f001:**
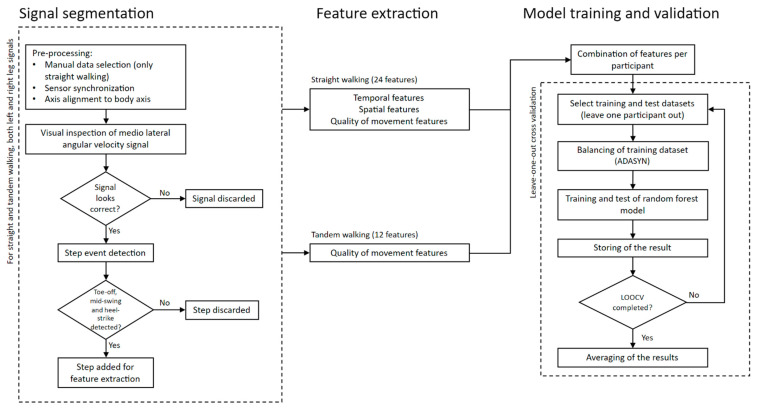
Flowchart of the entire analysis procedure, including signal segmentation, feature extraction and model training and validation.

**Figure 2 sensors-23-08410-f002:**
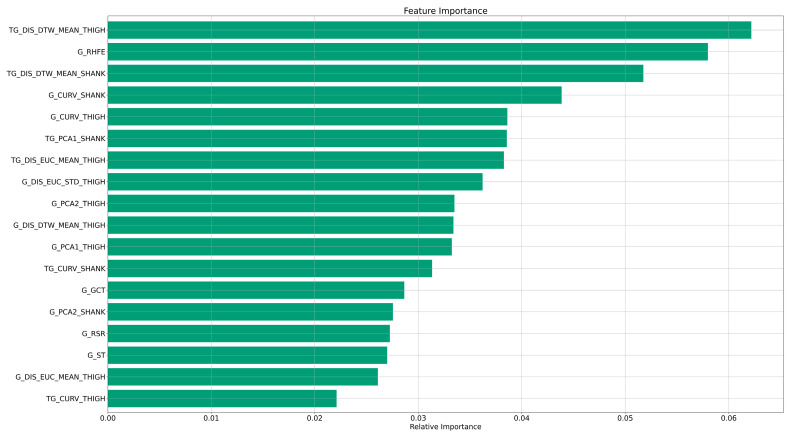
Average relative feature importance of the 20 most relevant features for classification. The feature importance according to the decrease of node impurity is normalised across all features. For details and explanation of all features, see Appendix A.

**Figure 3 sensors-23-08410-f003:**
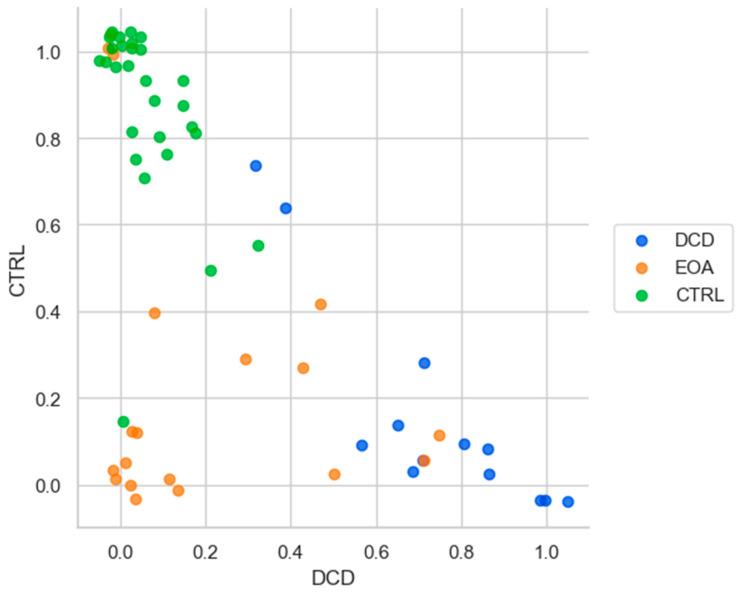
Mean probability over 100 iterations, each dot represents a participant from one of the three groups, EOA, DCD and healthy participants. EOA: early onset ataxia, DCD: developmental coordination disorder, CTRL: healthy control. The relative position of the participant indicates the estimated probability of being identified as EOA, DCD or CTRL. Dots near the (0,0) coordinate were identified as belonging to the EOA group, dots close to the (1,0) coordinate were identified as DCD patients and finally, dots in the (0,1) coordinate were classified as CTRL. The area outside these coordinates was defined as the intersection area, where each dot in this region is defined by its relative probability to have characteristics from the three groups (i.e., a dot at position (0.4,0.2) indicates that this participant has 0.4 probability of being classified as a DCD patient, 0.2 probability of being classified as a healthy participant and (1 − 0.4 − 0.2 = 0.4) probability of being classified as an EOA patient). We added jitter to the probability values to improve the visualisation of each participant and its relative position on the plot.

**Figure 4 sensors-23-08410-f004:**
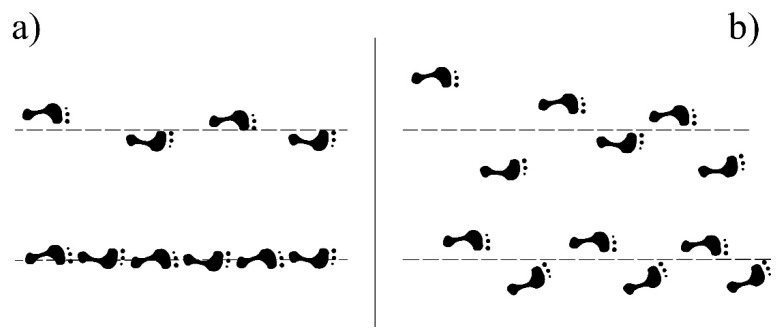
Typical clinically observed gait patterns in healthy persons and ataxia patients. (**a**) Gait (**top**) and tandem gait (**bottom**) of a healthy participant, (**b**) gait (**top**) and tandem gait (**bottom**) of an ataxic patient. Figure adapted with permission from Oosterhuis [26]. Copyright 1997, Oosterhuis.

**Table 1 sensors-23-08410-t001:** Participant characteristics for each of the three groups. EOA: early onset ataxia, DCD: developmental coordination disorder, CTRL: typically developing, IQR: interquartile range.

	EOA (n = 18)	DCD (n = 13)	CTRL (n = 29)
*Age (median, IQR)*	11.5 (8.5)	10.0 (4.5)	12.0 (6.0)
*Range (years)*	6–25	6–16	5–25
*Sex (male/female)*	11/7	8/5	11/18

**Table 2 sensors-23-08410-t002:** Confusion matrix averaged over 100 iterations and subsequently expressed in percentages. Each row displays the mean percentage of participants predicted to belong to each of the three classes. Overall mean accuracy is 82%. EOA: early onset ataxia, DCD: developmental coordination disorder, CTRL: healthy control. Note that the percentages do not add up to exactly 100% due to the LOOCV and the 100 iterations.

Prediction			
Actual	EOA	DCD	CTRL
EOA	61.4% (6.3%)	21.9% (6.2%)	16.6% (4.9%)
DCD	3.8% (4.7%)	82.5% (5.9%)	13.7% (3.8%)
CTRL	4.3% (1.5%)	1.2% (1.9%)	94.5% (2.2%)

**Table 3 sensors-23-08410-t003:** Measures of classification performance for EOA, CTRL and DCD participants as a positive class. EOA: early onset ataxia, DCD: developmental coordination disorder, CTRL: healthy control.

Actual	Balanced Accuracy	Sensitivity (Recall)	Specificity	Precision	F_1_-Score
EOA	0.79 (0.03)	0.61 (0.06)	0.96 (0.02)	0.87 (0.05)	0.72 (0.05)
DCD	0.87 (0.03)	0.82 (0.06)	0.91 (0.03)	0.72 (0.07)	0.77 (0.05)
CTRL	0.90 (0.02)	0.95 (0.02)	0.85 (0.03)	0.85 (0.03)	0.90 (0.02)

**Table 4 sensors-23-08410-t004:** Median values for the four most relevant gait characteristics and result of the Kruskal–Wallis test for each of the characteristics. EOA: early onset ataxia, DCD: developmental coordination disorder and CTRL: healthy control.

*Feature*	EOA	DCD	CTRL	Kruskal–Wallis Test
*TG_DIS_DTW_MEAN_THIGH*	578.42	532.75	212.43	χ^2^(2) = 100.11, *p* = 1.819 × 10^−22^
*G_RHFE*	203.64	183.91	173.80	χ^2^(2) = 45.51, *p* = 1.306 × 10^−10^
*TG_DIS_DTW_MEAN_SHANK*	631.38	597.52	200.56	χ^2^(2) = 103.68, *p* = 30.57 × 10^−23^
*G_CURV_SHANK*	0.0012	0.0017	0.0013	χ^2^(2) = 25.531, *p* = 2.857 × 10^−6^

## Data Availability

The data presented in this study are not available on request, as participants or their legal representatives did not provide consent.

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
