# Peer review of "Instrumented Gait Classification Using Meaningful Features in Patients with Impaired Coordination"

_sensors, 2023, doi:10.3390/s23208410_

Round 1

Reviewer 1 Report

This study aims to distinguish Early onset ataxia (EOA) and developmental coordination disorder (DCD) from each other and from healthy controls. The authors examine gait and tandem gait as captured by wearable movement sensors during performing the tests for the SARA ataxia score.

The authors use machine learning techniques for a multi-variate classification with the aim of identifying meaningful features for the classification of the three groups.

Understanding the development and the course of disease in EOA and DCD is an important and interesting research field. Unfortunately, I don’t think that the manuscript in this version is a valuable contribution to the field.

What the study does is throw a lot of expert-selected features into a machine learning approach that yields slightly better classification results than earlier work by the authors.  But what we do learn from this ?

-          For instance, a very interesting question would be, which movements have been mis-classified ? Are these subjects patients with a mild or early form of EOA ? Or played age or height a role ? What are the SARA scores of the mis-classified subjects ? Is the mis-classification caused by the walk or the tandem walk ?  How is the SARA and the SARA gait item for the three groups ?  

-          Which subjects showed changes in normal walk and which only in tandem ?

-          -> there would be a lot of interesting questions for understanding beyond the pure classification   

Other points:

-          The authors analyzed gait recorded from  the SARA gait items consisting of six meters gait turn and six meters back. This path including gait initiation, preparing for turn, and gait termination is clearly not adequate for calculating gait variability measures. All the variability measures will be strongly influenced by height and cognitive influences like when to prepare the turn and when to terminate the walk.  Only the average parameters may have an adequate reliability.

-          Kandel, Neuroscience is sure not the adequate reference for explaining EOA

-          The authors identified the range of thigh rotation in the sagittal plane for gait, and movement variability for tandem gait.  

o   In which sense are these features meaningful ? In current research in the field of ataxia , gait measures have to have a clear relevance for the patient to be meaningful. How is thigh rotation in the sagittal plane meaningful for gait ? And in how far is tigh rotation influenced by the age and the height of the subject ?

Author Response

Dear reviewer,

in the attached document we provide responses to your and the other two reviewers' comments.

Kind regards, on behalf of all co-authors,

Natasha Maurits.

Reviewer 2 Report

Comments are provided in the attached file. 

minor revision required

Author Response

(The authors gave the same response as above.)

Reviewer 3 Report

The authors proposed an instrumented gait classification using gait features to distinguish between early onset ataxia (EOA), developmental coordination disorder (DCD), and healthy controls. Overall this article is good. However, I have the following concerns:

1. In order to allow other researchers to quickly understand the research value of this paper, the authors should justify the importance of distinguishing EOA, DCD, and healthy controls in the introduction. Moreover, the authors should introduce methods used in clinical practice (if existed) and give their accuracy in the real-world usage.

2. The authors chose [10, 12, 16, 17] as related works. However, those works are somehow too old, among which the latest one was published in 2017. I wonder are there any state-of-the-art studies about distinguishing EOA, DCD, and healthy controls using gait or other tests? If not, what are the potential reasons for the paucity of relevant research? Meanwhile, the authors should give the accuracy of [10,12,17] in Line 348 and the corresponding accuracy of the proposed classification to compare with those works focusing on distinguishing DCD from healthy controls.

3. As described in Line 207, ten instances were used to obtain the label of patient based on majority vote strategy. However, how to determine the label when the number of votes is tied. Furthermore, it is vague that whether the Table 2 is the result of instances or patients? Nevertheless, both instance and patient results should be presented.

4. Figure 1 should be replaced by a high definition version.

Author Response

(The authors gave the same response as above.)

Round 2

Reviewer 2 Report

Author of the manuscript has addressed all the  comments that are given in the first round of the review.   

Reviewer 3 Report

The authors have carefully revised the entire manuscript and I am satisfied how the authors fulfilled previous requirements. I am suggesting manuscript acceptance.